# Modular total syntheses of *trans*-clerodanes and sesquiterpene (hydro)quinones via tail-to-head cyclization and reductive coupling strategies

Wenming Zhu[1,2], Qishuang Yin[1,2], Zhizheng Lou[1] & Ming Yang [1]✉

The *trans*-clerodanes and sesquiterpene (hydro)quinones are a growing class of natural products that exhibit a wide range of biological activities. Although they are different types of natural products, some of them feature the same *trans*-decalin core structure. Here, we report the total syntheses of two members of *trans*-clerodanes, five members of sesquiterpene (hydro)quinones as well as the proposed structure of dysidavarone D via a modular synthetic route. A bioinspired tail-to-head cyclization strategy was developed to syntheses of the *trans*-decalin architectures by using two diastereochemically complementary radical polyene cyclization reactions catalyzed by Ti(III) and mediated by Mn(III), respectively. The different types of side chains were introduced by challenging nickel catalyzed reductive couplings of sterically hindered alkyl halides. The synthesis of the proposed dysidavarone D proved a wrong structural assignment of the natural product.

The *trans*-clerodanes and sesquiterpene (hydro)quinones are different types of natural products, both of which exhibit a wide range of biological activities[1–3]. However, some of them (2–10, Fig. 1a) feature the same *trans*-decalin core structure (1 or *ent*-1, Fig. 1a)[4–26]. It suggests that these might have a similar biosynthetic pathway in Nature. As shown in Fig. 1b, the *trans*-decalin skeleton was proposed to generate by cyclases catalyzed proton-initiated cationic polyene cyclization followed by a series of Wagner-Meerwein rearrangements[1,2]. The biosynthetic pathway could be divided into two stages–the head-to-tail cyclization stage (11–12) and the domino rearrangement stage (12–14). The *trans*-decalin structure (highlighted in blue) of the cyclization intermediate 12, if redrawn in alternative orientation, is similar to the domino rearrangement product 14. The only difference is the position of the methyl groups (highlighted in red). We envisioned whether 12 could be transformed into 14 by a "single migration". Titanium(III) mediated radical cyclization of epoxy alkenes was originally developed by RajanBabu and

Nugent[27] which was then expanded to radical polyene cyclization and extensively used in the total synthesis of natural products[28]. Comparison of the structure of 14 and 16−the titanium(III) mediated radical polyene cyclization product of the derivative of farnesol 15[29–32], both of them also have similar *trans*-decalin architecture (in blue). If the acetate and the methylene (in red) in 16, which were generated from the acetate and methyl group (the red part) in 15, were removed and the hydroxyl group was transformed into a methyl group, the *trans*-decalin architecture of *trans*-clerodanes and sesquiterpene (hydro)quinones could be delivered. We think that if the acetate and methyl group in 15 are pre-removed, the desired cyclization product may be obtained without the redundant acetate and methylene group. With this thought in mind, we designed the cyclization precursor 17. After titanium(III)-mediated radical polyene cyclization and introduction of the desired methyl group from the secondary hydroxyl group, the previously mentioned "single migration" to construct the *trans*-decalin core structure of *trans*-cler-

[1]State Key Laboratory of Applied Organic Chemistry and College of Chemistry and Chemical Engineering, Lanzhou University, 222 South Tianshui Road, 730000 Lanzhou, Gansu Province, China. [2]These authors contributed equally: Wenming Zhu, Qishuang Yin. ✉e-mail: yangming@lzu.edu.cn

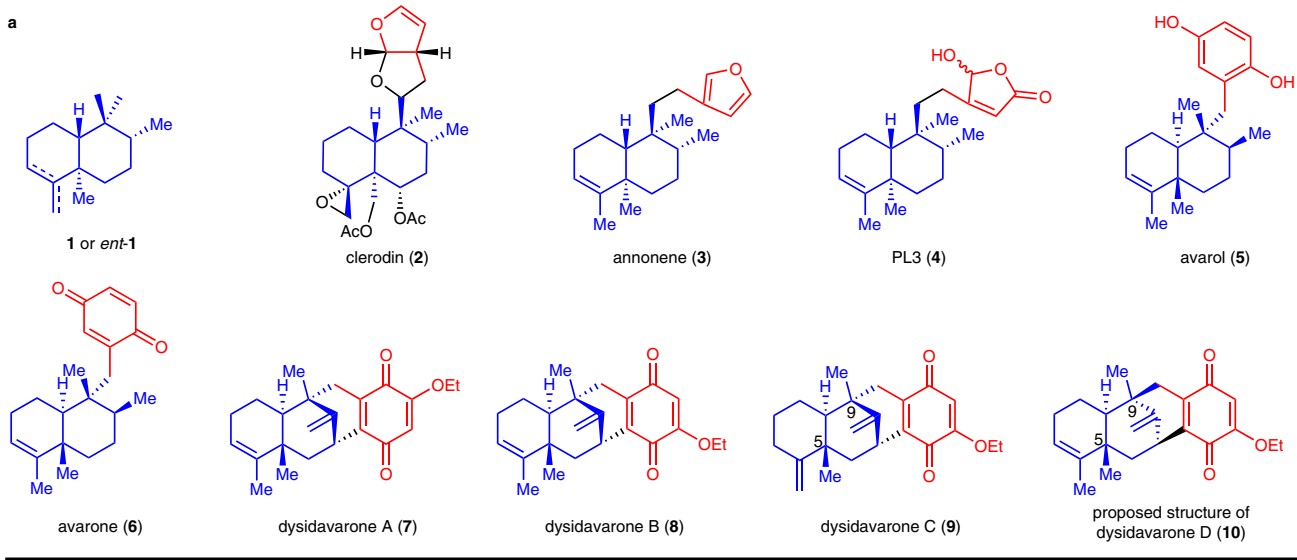

**Fig. 1 | Selected *trans*-clerodanes and sesquiterpene (hydo)quinones, the hypothesis of the biogenesis of such structures and our synthetic strategies.** **a** Selected *trans*-clerodanes and sesquiterpene (hydro)quinones that feature the similar *trans*-decalin core structure **1** or *ent*-**1**. **b** Nature's head-to-tail cyclization and domino rearrangement strategies to synthesize *trans*-clerodanes and some sesquiterpene (hydro)quinones. Our's tail-to-head cyclization and reductive coupling strategies to synthesize these types of natural products.

odanes and sesquiterpene (hydro)quinones could be achieved– this is a key differentiation from biosynthetic pathway where the cyclization of **17** initiated from the tail of the natural products and ended at the head. At last, the different types of side chains could be installed by nickel-catalyzed sterically hindered reductive couplings[33–35] that enables the modular route to this type of natural product. Herein, we report the total syntheses of **3**–**10** based on the tail-to-head cyclization and the reductive coupling strategies.

**Fig. 2 | Syntheses of annonene and PL3.** Reagents and conditions: **a** (D)-DET (0.1 equiv), Ti(Oi-Pr)₄ (0.05 equiv), TBHP (1.5 equiv, 5.5 M in decane), 3 Å MS, CH₂Cl₂, −30 °C, 12 h, then TBSCl (2.0 equiv), imidazole (3.0 equiv), 0 °C, 1 h, 93%; **b** 21 (1.5 equiv), Li₂CuCl₄ (0.1 equiv), THF, 0 °C, 1 h, then K₂CO₃ (10.0 equiv), MeOH:THF = 1.5:1, 23 °C, 12 h, 72%; **c** Cp₂TiCl₂ (0.2 equiv), Zn (3.0 equiv), 2,4,6-collidine·HCl (2.5 equiv), THF, 40 °C, 3 h, 65%; **d** TPAP (0.1 equiv), NMO (2.0 equiv), 4 Å MS, CH₂Cl₂, 23 °C, 1 h, 82%; **e** NaHMDS (1.5 eqiv), Ph₃PCH₃Br (1.75 equiv), THF, 60 °C, 8 h, then TBAF (5.0 equiv), 4 h, 60%; **f** [Ir(cod)(Py)(PCy₃)]⁺PF₆⁻ (0.005 equiv), H₂, CH₂Cl₂, 23 °C, 2 h, then Co(Sal^{t-Bu,t-Bu})Cl (0.05 equiv), PhSiH₃ (0.2 equiv), benzene, 23 °C, 8 h, 73%; **g** CBr₄ (2.0 equiv), PPh₃ (3.0 equiv), Py, 60 °C, 4 h, 53%; **h** 24 (3.0 equiv), NiBr₂ (0.3 equiv), Mn (4.0 equiv), P(4-CF₃Ph)₃ (0.3 equiv), DMF:DMSO = 1:1, 60 °C, 16 h, 50%; **i** Rose·Bengal (0.025 equiv), DIPEA (10.0 equiv), tungsten lamp (200 W), CH₂Cl₂, −78 °C, 3 h, 78%. (D)-DET (−)-diethyl D-tartrate, TBHP t-butyl hydroperoxide, MS molecular seives, TBSCl t-butyldimethylsilyl chloride, THF tetrahydrofuran, TPAP tetrapropylammonium perruthenate, NMO 4-methylmorpholine N-oxide, NaHMDS sodium bis(trimethylsilyl)amide, DMF N,N-dimethylformamide; DMSO dimethyl sulfoxide, TBAF tetrabutylammonium fluoride, DIPEA N-ethyldiisopropylamine.

## Results

### The syntheses of annonene and PL-3

We first investigated our strategies in the total synthesis of *trans*-clerodanes (Fig. 2). Starting from the allylic alcohol **19** (one step from geranyl acetate), Sharpless epoxidation with (−)-diethyl D-tartrate ((D)-DET), Ti(Oi-Pr)₄ and t-butyl hydroperoxide (TBHP) introduced the initial chiral center followed by one pot protection of the alcohol with t-butyldimethylsilyl chloride (TBSCl) afforded the epoxide **20** in 93% yield. **20** was subjected to a copper (Li₂CuCl₄) catalyzed coupling reaction with Grignard reagent **21** to assemble the alkyne chain, and subsequent one-pot deprotection of the trimethylsilyl (TMS) with K₂CO₃ and methanol (MeOH) to deliver cyclization precursor **17** in 72% yield[36]. With **17** in hand, the titanium(III) catalyzed radical polyene cyclization was investigated. Previous research indicated that this type of radical polyene cyclization could be achieved by using two equivalents of Cp₂TiCl at 60 °C, yet

**Fig. 3 | Syntheses of avarone and avarol.** Reagents and conditions: **a** I₂ (2.0 equiv), Imidazole (3.0 equiv), PPh₃ (3.0 equiv), benzene, 50 °C, 12 h, 61%; **b** 27 (3.0 equiv), NiI₂ (0.2 equiv), 28 (0.2 equiv), Zn (3.0 equiv), DMAc, 80 °C, 2 h, 63%; **c** salcomine (2.5 equiv), O₂, DMF, 40 °C, 2 days, 71%; **d** salcomine (2.5 equiv), O₂, DMF, 40 °C, 2 days, then aqueous Na₂S₂O₄, 63%. DMAc dimethylacetamide.

no catalytic reaction was reported[37]. Inspired by Gansäuer's work[38], and Nugent and RajanBabu's recent essay[39], we found that the radical polyene cyclization could proceed smoothly to generate **18** in 65% yield when treatment of **17** with 0.2 equiv of Cp₂TiCl₂, 2.5 equiv of 2,4,6-collidine·HCl and 3.0 equiv of Zn at 40 °C. The 2,4,6-collidine·HCl played important roles in the catalytic reaction. Not only as the proton source to regenerate the Cp₂TiCl₂ from the intermediate, but 2,4,6-collidine·HCl could also interact with Cp₂TiCl to form a complex which could decrease the transient concentration of free Cp₂TiCl[39–41]. This effect could arrest the disproportionation of the initial radical whose rate is dependent on the concentration of Cp₂TiCl. Pressing forward, exposure of **18** to Ley's oxidation conditions enabled the oxidation of secondary alcohol to a ketone. Subsequent Wittig reaction of the resultant ketone followed by one-pot deprotection of the t-butyldimethylsilyl (TBS) with tetrabutylammonium fluoride (TBAF) gave alcohol **22** in 49% yield over 2 steps. Next, in anticipation of probing the projected reductive coupling, alcohol **22** was transformed into bromide **23** in 39% overall yield through hydroxyl group directed hydrogenation of one of the terminal alkenes with Crabtree's catalyst[42,43] and one-pot isomerization of another terminal alkene with Co(Sal^{t-Bu, t-Bu})Cl and PhSiH₃[44], followed by bromination of the hydroxyl group with CBr₄ and Ph₃P. Pleasingly, we found that the desired reductive coupling of **23** with **24** to generate annonene (**3**) could indeed be effected using Shu's conditions [NiBr₂, Mn, (4-CF₃Ph)₃P][45]. Photosensitized oxidation of **3** [O₂, Rose Bengal, N,N-diisopropylethylamine (DIPEA), visible light] gave 16-hydroxycleroda-3,13-dien-15,16-olide (PL-3, **4**) in 78% yield through the [4 + 2] addition of singlet oxygen to the 3-alkylfuran and regiospecific deprotonation[46,47].

### The syntheses of avarone and avarol

Having successfully completed the total syntheses of annonene and avarol (Fig. 3). Compound **25** was synthesized from geranyl acetate in a similar fashion to that described above in 7 steps in 11% overall yield. Iodination of the hydroxyl group with Ph₃P and I₂ afforded iodide **26** in 61% yield. NiI₂ catalyzed reductive coupling of **26** and 3-bromo-phenol (**27**) gave **29** in 63% yield[48–51]. Lastly, oxidation of the resultant phenol with salcomine and O₂ provided avarone (**6**) in 71% yield[52,53], while quenching with an aqueous solution of Na₂S₂O₄ led to avarol (**5**) in 63% yield.

**Fig. 4 | Syntheses of dysidavarones A–C.** Reagents and conditions: **a** HF·Py:THF = 1:4, 40 °C, 2 h, 73%; **b** I₂ (2.0 equiv), PPh₃ (3.0 equiv), benzene, 23 °C, 16 h, 89%; **c** 32 (3.0 equiv), **28** (0.2 equiv), NiBr₂.DME (0.2 equiv), Mn (3.0 equiv), DMAc, 23 °C, 16 h; **d** NHC-Pd(II)-Im (0.2 equiv), *t*-BuONa (3.0 equiv), 1,4-dioxane, 110 °C, 3 h, 38% (2 steps); **e** CH₃PPh₃Br (10.0 equiv), *t*-BuOK (10.0 equiv), toluene, 120 °C, 7 h, 88%; **f** *n*-BuSH (10.0 equiv), *n*-BuLi (10.0 equiv), HMPA, 110 °C, 4 h, 85%; **g** salcomine (3.0 equiv), O₂, acetonitrile, 23 °C, 85%; **h** Et₃N (50.0 equiv), O₂, EtOH, 30 °C, **9**, 19%, **35**, 25%; **i** PTSA·H₂O (0.2 equiv), HOAc, 40 °C, **8**, 54%, **7**, 63%. DME 1,2-dimethoxyethane, HMPA hexamethylphosphoramide, PTSA·H₂O *p*-toluenesulfonic acid monohydrate.

### The syntheses of dysidavarones A, B and C

The strategies were then applied to the syntheses of dysidavarones A, B and C (**7–9**) which possess tetracyclic frameworks (Fig. 4). Compound **30** could also be synthesized from geranyl acetate in a similar fashion to that described above in 5 steps in 25% overall yield. Deprotection of TBS ether with HF·py followed by iodination of the resultant hydroxyl group afforded the iodide **31** in 65% yield over two steps. The reductive coupling of **31** and **32** was achieved under the NiBr₂·DME catalyzed conditions[48–51]. Subsequent palladium [NHC-Pd(II)-Im] catalyzed intramolecular α-arylation of ketone to assemble the bicyclo[3.3.1] architecture provided **33** in 38% overall yield[54]. The stereochemistry of **33** was unambiguously confirmed by the X-ray crystallographic analysis. Compound **33** was then converted into quinone **34** in 64% overall yield through Wittig reaction and removal of methyl protection group by *n*-BuSLi[55], followed by the oxidation of the resultant phenol to quinone with salcomine and O₂[52,53]. The ethoxy group in dysidavarones was introduced by treating **34** with Et₃N in ethanol (EtOH) under O₂[56,57], affording dysidavarone C (**9**) and **35** in 19% and 25% yield, respectively. Isomerization of terminal alkene in **9** under acidic conditions [*p*-toluenesulfonic acid monohydrate (PTSA·H₂O), acetic acid (HOAc)] provided dysidavarone B (**8**) in 54% yield[25]. Dysidavarone A (**7**) could also be obtained from **35** under the same conditions in 63% yield.

### The synthesis of proposed structure of dysidavarone D

Dysidavarone D also features a tetracyclic framework similar to dysidavarones A–C, however, the stereochemistry is different which methyl groups at C5 and C9 position of the *trans*-decalin architecture are *trans*-form. Unfortunately, efforts to construct the *trans*-decalin architecture of dysidavarone D via titanium(III) catalyzed radical polyene cyclization were unfruitful[58]. As a result, we developed an alternate tail-to-head cyclization route to synthesize dysidavarone D by a Mn(OAc)₃·2H₂O mediated radical polyene cyclization (Fig. 5)[59]. Allylation of the bisenolate of **37** with allyl bromide **36** delivered **38** (73% yield), which was subjected to the Mn(OAc)₃·2H₂O mediated radical polyene cyclization conditions to give keto ester **39** (61% yield). The keto ester **39** was then converted into iodide **40** through the protection of the ketone by transforming it into enol methyl ether, LiAlH₄ reduction of the ester to primary alcohol followed by one-pot hydrolysis of methyl enol ether to ketone, and iodination of the resultant hydroxyl group. Due to the *cis* configuration of the C5-methyl group and the C9-iodomethyl group, the reductive coupling of iodide **40** and **32** would be more challenging than previous substrates. Catalytic conditions only afforded a trace amount of coupling product. After extensive optimization, we found that 1.5 equiv of NiBr₂·DME and ligand **28** could promote the desired reductive coupling with acceptable efficiency. Subsequent isomerization of the terminal alkene delivered **41** in 21% overall yield. Quinone **42** was synthesized from **41** in a similar fashion to that described above in the syntheses of dysidavarones A–C in 4 steps. The stereochemistry of the tetracyclic framework was unambiguously confirmed by the X-ray crystallographic analysis of the intramolecular α-arylation product of ketone **41** (see Supplementary Information). The introduction of the ethoxy group was achieved by treating **42** with Co(OAc)₂ in EtOH under O₂, delivering a 1:1 mixture of inseparable regioisomers in 42% yield (70% yield based on recovered starting material)[60]. The resulting quinone mixtures were reduced to hydroquinones, followed by selective mono-TBS-protection generated silyl ethers **43** and **44** in 37% and 26% yield, respectively, which were separable at this stage. The position of ethoxy groups in **43** and **44** was unambiguously confirmed by its nuclear Overhauser effect spectroscopy (NOESY) spectra. Lastly, **43** and **44** were converted into corresponding quinones **10** and **45** in 87% and 89% yield, respectively, through deprotection of TBS with HF·py following oxidation of the resultant hydroquinones to quinones with MnO₂[61]. However, neither NMR spectra of **10** nor **45** matched with the originally proposed structure of dysidavarone D, suggesting the original structural assignment may be incorrect[15].

## Discussion

In conclusion, we have accomplished the total syntheses of a number of *trans*-clerodanes and sesquiterpene (hydro)quinones through tail-to-head cyclization and reductive coupling strategies. The synthesis of the proposed dysidavarone D also suggested incorrect structural assignments from the previous report. We believe that the flexible, modular, and generalized biomimetic[62] synthetic route may facilitate biological studies of these natural products and their derivatives.

## Methods

All reactions were carried out under an argon atmosphere with dry solvents under anhydrous conditions unless otherwise noted. Tetrahydrofuran (THF), diethyl ether (Et₂O), and toluene were distilled immediately before use from sodium-benzophenone ketyl. Dimethylformamide (DMF) and dichloromethane (CH₂Cl₂) were distilled from calcium hydride and stored under an argon atmosphere. Yields refer to chromatographically and spectroscopically (¹H and ¹³C NMR) homogeneous materials unless otherwise stated. Reagents were purchased at the highest commercial quality and used without further purification unless otherwise stated. Reactions were magnetically stirred and monitored by thin-layer chromatography (TLC) carried out on

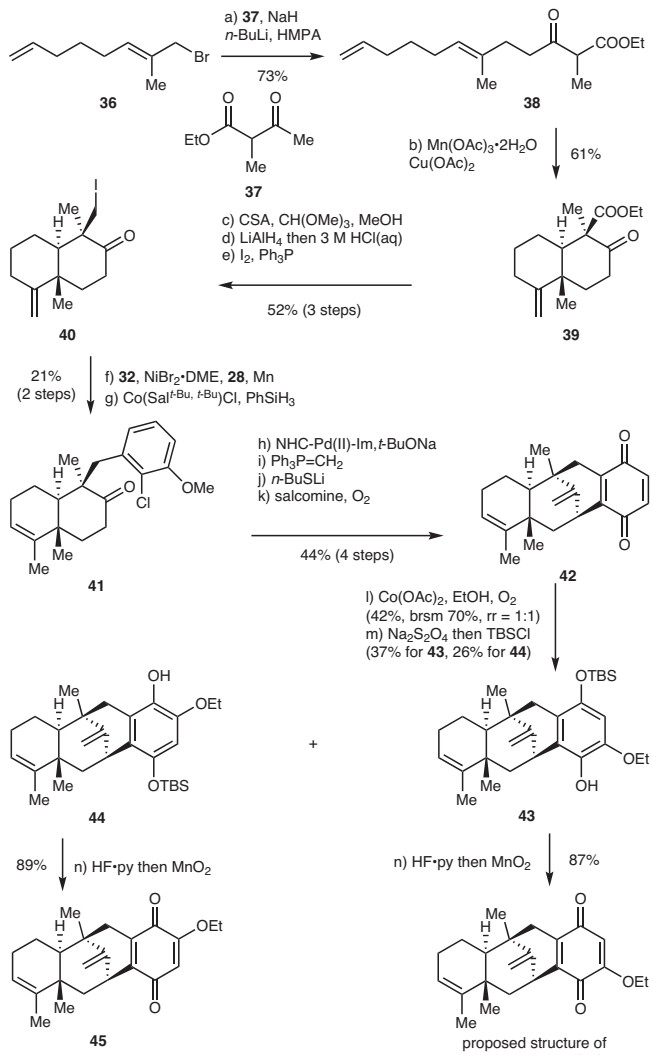

**Fig. 5 | Synthesis of proposed structure of dysidavarone D.** Reagents and conditions: **a** NaH (5.0 equiv), *n*-BuLi (5.0 equiv), **37** (5.0 equiv), HMPA (1.2 equiv), THF, 23 °C, 1 h, 73%; **b** Mn(OAc)$_3$·2H$_2$O (2.0 equiv), Cu(OAc)$_2$ (1.1 equiv), AcOH, 40 °C, 24 h, 61%; **c** CH(OMe)$_3$ (7.5 equiv), CSA (0.10 equiv), MeOH, 23 °C, 4.5 h, 81%; **d** LiAlH$_4$ (4.0 equiv), THF, 15 h, then 3 M HCl, 23 °C, 1 h, 75%; **e** PPh$_3$ (1.3 equiv), I$_2$ (1.2 equiv), imidazole (2.0 equiv), benzene, 40 °C, 15 h, 85%; **f** NiBr·DME (1.5 equiv), **28** (1.5 equiv), Mn (3.0 equiv), **32** (3.0 equiv), DMAc, 17 h; **g** Co(Sal$^{t\text{-}Bu, t\text{-}Bu}$)Cl (0.05 equiv), PhSiH$_3$ (0.2 equiv), benzene, 23 °C, 24 h, 21% (2 steps); **h** NHC-Pd(II)-Im (0.1 equiv), *t*-BuONa (3.0 equiv), 1,4-dioxane, 110 °C, 6 h, 65%; **i** *t*-BuOK (10.0 equiv) PPh$_3$CH$_3$Br (10.0 equiv), toluene, 120 °C, 8 h, 91%; **j** *n*-BuSH (7.5 equiv), *n*-BuLi (7.5 equiv), HMPA, 110 °C, 89%; **k** salcomine (3.0 equiv), O$_2$, acetonitrile, 23 °C, 3 h, 83%; **l** Co(OAc)$_2$ (3.0 equiv), O$_2$, EtOH, 160 °C, 10 h, 42% (b.r.m. 70%, r.r. = 1:1); **m** aqueous Na$_2$S$_2$O$_4$, THF, 1 h, then imidazole (5.0 equiv), TBSCl (3.0 equiv), DCM, −10 °C, 37% for **43**, 26% for **44**; and **n** HF·Py:THF = 1:2, 23 °C, 1 h, then MnO$_2$, CH$_2$Cl$_2$, 23 °C, 15 min, 87% for proposed structure of dysidavarone D (**10**), 89% for **45**.

0.25 mm Xinnuo silica gel plates (60F-254) using UV light as a visualizing agent, and an ethanolic solution of phosphomolybdic acid and cerium sulfate, and heat as developing agents. Steema silica gel (60, academic grade, particle size 0.040–0.063 mm) was used for flash column chromatography. Preparative thin-layer chromatography separations were carried out on 0.50 mm Xinuo silica gel plates (60F-254). NMR spectra were recorded on Bruker 600 MHz and 400 MHz instruments and calibrated using residual undeuterated solvent as an internal reference. The following abbreviations were used to explain the multiplicities: s = singlet, d = doublet, t = triplet, q = quartet, m = multiplet. IR spectra were recorded on a Perkin-Elmer 1000 series FT-

IR spectrometer. High-resolution mass spectra (HRMS) were recorded on an Agilent 6244 Tof-MS using ESI (Electrospray Ionization).

## Data availability
The data generated in this study are provided in the Supplementary Information file. The experimental procedures, copies of all spectra data and full characterization have been deposited in Supplementary Information file.

## Code availability
The X-ray crystallographic coordinates for structures (**33** and **SI-12**) reported in this Article have been deposited at the Cambridge Crystallographic Data Centre (CCDC), under deposition number CCDC 2184139 and 2205566. These data can be obtained free of charge from The Cambridge Crystallographic Data Centre via www.ccdc.cam.ac.uk/data_request/cif.

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

## Acknowledgements

We thank Dr. Alison X. Gao (Novartis Pharmaceuticals) for the valuable discussion. We are grateful to Yongliang Shao for X-ray crystallographic analysis and Fengming Qi for NMR assistance and Hongli Chen for mass spectrometry assistance. Financial Support was provided by the Lanzhou University, National Natural Science Foundation of China (21901094 and 22071089, to M.Y.) and the Fundamental Research Funds for the Central Universities (lzujbky-2019-49 and lzujbky-2020-ct01, to M.Y.).

## Author contributions

M.Y., W.Z., and Q.Y conceived the synthetic route. M.Y. directed the project. W.Z. completed the synthesis of compounds **3**–**6** and **10**. Q.Y. completed the synthesis of compounds **7**–**9**. Z.L. participated in material preparation and explored various alternative pathways toward the target molecules. M.Y., W.Z., Q.Y., and Z.L. analyzed the results. M.Y. wrote the manuscript.

## Competing interests

The authors declare no competing interests.
