## [Peer Review File · Nature Communications]

Modular total syntheses of trans-clerodanes and sesquiterpene (hydro)quinones via tail-to-head cyclization and reductive coupling strategiesReviewers' Comments:

Reviewer #1:

Remarks to the Author:

This manuscript, by Yang and collaborators, reports the total syntheses of three trans-clerodanes, namely annonene, PL3 and avarol, and five other sesquiterpene (hydro)quinines, including the proposed structure of dysidavarone D. The synthetic routes lead to the target molecules via trans-decalin intermediates 18 or ent-18, and both of them were obtained by radical polyene cyclization reactions mediated with Nugent reagent. Although seven natural products have been synthesized, the natural products made are quite similar to one another with moderate molecular complexity. To the best of my knowledge, several natural products (annonene, avarol, avarone, dysidavarone A and C) presented in this manuscript have been made previously, and similar cyclization strategies to construct the trans-decalin core have been reported several times, for example in Cuerva's synthesis of terpenoids (Titanocene-Catalyzed Cascade Cyclization of Epoxypolyprenes: Straightforward Synthesis of Terpenoids by Free-Radical Chemistry, *Chem. Eur. J.* 2004, 10, 1778 - 1788) and Takahashi's synthesis of ent-pyripyropene A (Asymmetric Total Synthesis of ent-Pyripyropene A, *Chem. Eur. J.* 2015, 21, 9454-9460) using titanocene-catalyzed cyclization of enantiomerically pure epoxy alkenes. Although the chemistry in this manuscript is very fine synthetic work, for instance, racemic radical polyene cyclization mediated by Mn(III), and nickel catalyzed reductive couplings of sterically hindered alkyl halides, the moderate molecular complexity of these natural products in combination with the fact that they have previously been made several times, it falls short for publication in Nature Communication for reasons mentioned above.

Minor point: For dysidavarone D, it would be better to propose a structure based on data collected in this total synthesis and the original data reported in the previous structural assignment.

Tables for data comparison of synthesized compounds and the natural products should be provided in SI.

To make it clear, a step-by-step reaction Scheme for preparing compounds 25 and 30 from geranyl acetate should be provided in SI as well. Beside reaction steps, overall yields for intermediates 25 and 30 (Figure 3 and Figure 4) should be mentioned in the manuscript as well.

The references section, refs 7, 15, 18, 19, 22, 28, 29, 32, 40, 41, 58 and 62 should be revised according to the formatting requirement.

Reviewer #2:

Remarks to the Author:

A feature of this work lies in the modular construction of five key trans-decalin cores via catalytic Ti(III) or Mn(III)-mediated radical ring cyclization of enynes and dienes that are initiated from the epoxide and 1,3-dicarbonyl moieties. The resultant primary hydroxyl groups were converted to alkyl halides followed by Ni-catalyzed reductive coupling with furylmethylenyl oxalate and aryl halides. Subsequent oxidation and/or cyclization and/or alkene isomerization furnished the five final products. The authors also pointed out the structure of dysidavarone D assigned in the original paper was incorrect. Given the concise strategy utilized in this work for effective total syntheses of complex natural products containing up to four fused rings including quinone subunits, I am in support of publication of this work in Nature Commun with a few minor issues for the authors to address prior to its acceptance.

(1) The miss-assigned structure of dysidavarone D should be detailed in the SI by pinpointing the characteristic NMR peaks of the reported and the present ones. Also, in text, the authors should cite which paper they refer to as the original structural assignment for dysidavarone D.

(2) The use of the state-of-art reductive coupling protocol to install furylmethylenyl and aryl moieties is interesting. However, in some cases (e.g., 38 and 41), the yields are low. Are the authors aware of recent studies on formal alcohol cross-electrophile couplings by MacMillan, Gong and Weix? Perhaps it

is worth testing how it works for 38 or 41 with one of the new procedures.

(3) "dysidavarone D" in the legend of Figure 5 is misspelled.

(4) The radical cyclization appears to be highly diastereoselective. Can the authors provide a general model to predict the diastereoselective outcomes?

Reviewer #3:

Remarks to the Author:

Modular total syntheses of trans-clerodanes and sesquiterpene (hydro)quinones via tail-to-head cyclization and reductive coupling strategies

By Wenming Zhu, Qishuang Yin, Zhizheng Lou and Ming Yang

The authors describe a very nice approach to the synthesis of a series of trans decalin natural products based on a TiCp₂Cl reductive cyclization. The strategy is vaguely biomimetic, but the central cyclization is radical mediated rather than cationic. Their Ti system is catalytic, which is a feature. The Nickel mediated cross-electrophile coupling (e.g. 23 to 3) works well and a similar step is used in each synthesis. They extend this basic strategy to quinones and hydroquinone natural products avarone and avarol (Figure 3), and the approach is effective. They further extend the strategy to dysdavarone natural products, which are tetracyclic. The final ring is closed using a Pd cyclization of a ketone enolate onto an aryl bromide, which works in reasonable yield and is a very good application of Pd cyclization chemistry. They find that dysdavarone D, which requires a slightly different radical cyclization early on, does not match the reported literature. The proposed structure is most likely incorrect.

The paper describes a very interesting approach to the synthesis of this class of sesquiterpene natural products. However, I do not see any aspects that would appeal to scientists outside the specialty area of total synthesis. I think it is a great paper, but it would be more suitable for a journal that caters to chemistry.

I suggest adding a table in the SI to directly compare the proton and carbon spectra of their synthetic dysdavarone D with the data reported for the natural product. They simply state that it does not match, but they give no details and they should.

There are some minor oddities in phrasing, but overall it reads well. The manuscript is clean and I have no corrections.

Response to Referees

Referee #1

1. *This manuscript, by Yang and collaborators, reports the total syntheses of three trans-clerodanes, namely annonene, PL3 and avarol, and five other sesquiterpene (hydro)quinines, including the proposed structure of dysidavarone D. The synthetic routes lead to the target molecules via trans-decalin intermediates 18 or ent-18, and both of them were obtained by radical polyene cyclization reactions mediated with Nugent reagent. Although seven natural products have been synthesized, the natural products made are quite similar to one another with moderate molecular complexity. To the best of my knowledge, several natural products (annonene, avarol, avarone, dysidavarone A and C) presented in this manuscript have been made previously, and similar cyclization strategies to construct the trans-decalin core have been reported several times, for example in Cuerva's synthesis of terpenoids (Titanocene-Catalyzed Cascade Cyclization of Epoxy polyenes: Straightforward Synthesis of Terpenoids by Free-Radical Chemistry, Chem. Eur. J. 2004, 10, 1778 - 1788) and Takahashi's synthesis of ent-pyripyropene A (Asymmetric Total Synthesis of ent-Pyripyropene A, Chem. Eur. J. 2015, 21, 9454–9460) using titanocene-catalyzed cyclization of enantiomerically pure epoxy alkenes. Although the chemistry in this manuscript is very fine synthetic work, for instance, racemic radical polyene cyclization mediated by Mn(III), and nickel catalyzed reductive couplings of sterically hindered alkyl halides, the moderate molecular complexity of these natural products in combination with the fact that they have previously been made several times, it falls short for publication in Nature Communication for reasons mentioned above.*

Although the referee consider that our work falls short for publication in *Nature Communications* due to the moderate molecular complexity, several members of our targets have been synthesized before and the titanium(III) catalyzed radical polyene cyclization have been used in other natural products synthesis, we do appreciate his/her comments in terms of appreciating the chemistry in this manuscript is very fine synthetic work. We do wish to note that synthetic strategies are also important in evaluate a total synthesis besides the molecular complexity. The tail-to-head cyclization strategy which is different from the Nature's head-to-tail cyclization strategy, combined with reductive coupling strategy make our modular synthetic routes more flexible that could be used to efficiently synthesize different types of *trans*-clerodanes and sesquiterpene (hydro)quinones as well as proposed structure of dysidavarone D. Although there are several members of our targets have been synthesized before. However, the proposed structure of dysidavarone D could not be synthesized through any previously reported routes due to its specific stereochemistry. Also, our develop of Ti(III) **catalyzed** radical polyene cyclization of epoxy alkenes with an **alkyne as termination** expanded the application scope of this type of reaction and made the synthesis easily to scale up. Furthermore, the successful application of nickel catalyzed sterically hindered **reductive coupling at late stage** in our synthesis would accelerate the application of this novel reaction in other total synthesis or prodrug synthesis

2. *Minor point: For dysidavarone D, it would be better to propose a structure based on data collected in this total synthesis and the original data reported in the previous structural assignment. Tables for data comparison of synthesized compounds and the natural products should be provided in SI.*

We thank the referee's suggestion and added the Tables for data comparison of synthesized compound and the natural product (dysidavarone D) in SI. However, we can't propose the correct structure of dysidavarone D based on the data we got.

3. *To make it clear, a step-by-step reaction Scheme for preparing compounds 25 and 30 from geranyl acetate should be provided in SI as well. Beside reaction steps, overall yields for intermediates 25 and 30 (Figure 3 and Figure 4) should be mentioned in the manuscript as well.*

We thank the referee's suggestion and added three Figures in SI which contained step-by-step reactions for all compounds. The overall yields for intermediates 25 and 30 were also added in the revised manuscript.

4. *The references section, refs 7, 15, 18, 19, 22, 28, 29, 32, 40, 41, 58 and 62 should be revised according to the formatting requirement.*

According to the formatting requirement, if the number of authors in reference exceeded than six, the first author should be given, followed by "et al.". So the refs 7, 15, 18, 19, 22, 28, 29, 32, 40, 41, 58 and 62 are correct.

Other changes: 1) We have added the X-ray crystal data for compound **SI-12** in SI which was deposited at the Cambridge Crystallographic Data Centre (CCDC), under deposition number CCDC 2205566. We also added "The stereochemistry of the tetracyclic framework was unambiguously confirmed by the X-ray crystallographic analysis of the intramolecular α -arylation product of ketone **41** (see supplementary information)." at appropriate position. 2) The yield of isomerization of terminal alkene in **9** and **35** should be 54% and 63%, respectively. We have corrected it.

Referee #2

1. *A feature of this work lies in the modular construction of five key trans-decalin cores via catalytic Ti(III) or Mn(III)-mediated radical ring cyclization of enynes and dienes that are initiated from the epoxide and 1,3-dicarbonyl moieties. The resultant primary hydroxyl groups were converted to alkyl halides followed by Ni-catalyzed reductive coupling with furylmethylenyl oxalate and aryl halides. Subsequent oxidation and/or cyclization and/or alkene isomerization furnished the five final products. The authors also pointed out the structure of dysidavarone D assigned in the original paper was incorrect. Given the concise strategy utilized in this work for effective total syntheses of complex natural products containing up to four fused rings including quinone subunits, I am in support of publication of this work in Nature Commun with a few*

minor issues for the authors to address prior to its acceptance.

We thank the referee for his/her comment and opinion on our work.

- (1) The miss-assigned structure of dysidavarone D should be detailed in the SI by pinpointing the characteristic NMR peaks of the reported and the present ones. Also, in text, the authors should cite which paper they refer to as the original structural assignment for dysidavarone D.*

We thank the referee's suggestion and added the Tables for data comparison of synthesized compound and the natural product (dysidavarone D) in SI. We also cited the isolation paper at appropriate position in the revised manuscript.

- (2) The use of the state-of-art reductive coupling protocol to install furylmethylenyl and aryl moieties is interesting. However, in some cases (e.g., 38 and 41), the yields are low. Are the authors aware of recent studies on formal alcohol cross-electrophile couplings by MacMillan, Gong and Weix? Perhaps it is worth testing how it works for 38 or 41 with one of the new procedures.*

We thank the referee's suggestion. The formal alcohol cross-electrophile couplings developed by MacMillan, Gong and Weix groups respectively are very interesting and useful reactions. However, when we tried MacMillan's conditions for the coupling of **SI-7** and **32**, only trace amount of coupling product was obtained. The major product is the ring expansion product which was formed via an intramolecular methyl radical addition to the ketone. We think that the intramolecular radical addition is faster than the intermolecular trapping with ArNi(II)Br species. In the reductive coupling of alkyl iodide and aryl halides that we used in the manuscript, the alkyl radical might be formed as ArNi(III)alkyl X species by the reaction with ArNi(II)X, which lead to the coupling products as major products. We also tried Gong's conditions which formed alkyl bromide in situ for the reductive coupling. However, no reaction was occurred. The Weix's conditions were also tried which cause the retro aldol reaction.

- (3) "dysidavaron D" in the legend of Figure 5 is misspelled.*

We thank the referee for his/her carefully checking our manuscript and changed the "dysidavaron D" to "dysidavarone D" in the legend of Figure 5. We also changed the "dysidavarons A-C" to "dysidavarones A-C" in the legend of Figure 4.

- (4) The radical cyclization appears to be highly diastereoselective. Can the authors provide a general model to predict the diastereoselective outcomes?*

According to the previously reported examples, both the titanium(III) catalyzed radical polyene cyclization and the Mn(OAc)₃·H₂O mediated radical polyene cyclization are stereospecific reactions. However the stereochemistry of the

products are different. The C1 hydroxymethyl group group in the product of titanium(III) catalyzed radical polyene cyclization is *trans* to the C5 methyl group. The C1 ester groups in the product of Mn(OAc)₃·H₂O mediated radical polyene cyclization is *cis* to the C5 methyl group. The transition states of the cyclization reactions are shown below. It should be noted that if the TBSOCH₂ is changed to ester group in the titanium(III) catalyzed radical polyene cyclization reaction, the stereochemistry will be the same.

Other changes: 1) We have added the X-ray crystal data for compound **SI-12** in SI which was deposited at the Cambridge Crystallographic Data Centre (CCDC), under deposition number CCDC 2205566. We also added “The stereochemistry of the tetracyclic framework was unambiguously confirmed by the X-ray crystallographic analysis of the intramolecular α -arylation product of ketone **41** (see supplementary information).” at appropriate position. 2) The yield of isomerization of terminal alkene in **9** and **35** should be 54% and 63%, respectively. We have corrected it.

Referee #3

- Modular total syntheses of trans-clerodanes and sesquiterpene (hydro)quinones via tail-to-head cyclization and reductive coupling strategies*
 By Wenming Zhu, Qishuang Yin, Zhizheng Lou and Ming Yang
 The authors describe a very nice approach to the synthesis of a series of trans decalin natural products based on a TiCp₂Cl reductive cyclization. The strategy is vaguely biomimetic, but the central cyclization is radical mediated rather than cationic. Their Ti system is catalytic, which is a feature. The Nickel mediated cross-electrophile coupling (e.g. 23 to 3) works well and a similar step is used in each synthesis. They extend this basic strategy to quinones and hydroquinone natural products avarone and avarol (Figure 3), and the approach is effective. They further extend the strategy to dysdavarone natural products, which are tetracyclic. The final ring is closed using a Pd cyclization of a ketone enolate onto an aryl bromide, which works in reasonable yield and is a very good application of Pd cyclization chemistry. They find that dysdavarone D, which requires a slightly different radical cyclization early on, does not match the reported literature. The proposed structure is most likely incorrect. The paper describes a very interesting approach to the synthesis of this class of sesquiterpene natural products. However, I do not see any aspects that would appeal to scientists outside the specialty area of total synthesis. I think it is a great paper, but it would be more suitable for a journal that caters to chemistry.

We thank the referee for his/her comment and opinion on our work.

- I suggest adding a table in the SI to directly compare the proton and carbon spectra of their synthetic dysdavarone D with the data reported for the natural product. They simply state that it does not match, but they give no details and they should. There are some minor oddities in phrasing, but overall it reads well. The manuscript is clean and I have no corrections.*

We thank the referee's suggestion and added the Tables for data comparison of synthesized compound and the natural product (dysidavarone D) in SI.

Other changes: 1) We have added the X-ray crystal data for compound **SI-12** in SI which was deposited at the Cambridge Crystallographic Data Centre (CCDC), under deposition number CCDC 2205566. We also added "The stereochemistry of the tetracyclic framework was unambiguously confirmed by the X-ray crystallographic analysis of the intramolecular α -arylation product of ketone **41** (see supplementary information)." at appropriate position. 2) The yield of isomerization of terminal alkene in **9** and **35** should be 54% and 63%, respectively. We have corrected it.

Reviewers' Comments:

Reviewer #2:

Remarks to the Author:

The authors have addressed my questions. I recommend publication of this work.